# Hyper-Graph-Network Decoders for Block Codes

**Eliya Nachmani and Lior Wolf**
Facebook AI Research and Tel Aviv University

## Abstract

Neural decoders were shown to outperform classical message passing techniques for short BCH codes. In this work, we extend these results to much larger families of algebraic block codes, by performing message passing with graph neural networks. The parameters of the sub-network at each variable-node in the Tanner graph are obtained from a hypernetwork that receives the absolute values of the current message as input. To add stability, we employ a simplified version of the arctanh activation that is based on a high order Taylor approximation of this activation function. Our results show that for a large number of algebraic block codes, from diverse families of codes (BCH, LDPC, Polar), the decoding obtained with our method outperforms the vanilla belief propagation method as well as other learning techniques from the literature.

## 1 Introduction

Decoding algebraic block codes is an open problem and learning techniques have recently been introduced to this field. While the first networks were fully connected (FC) networks, these were replaced with recurrent neural networks (RNNs), which follow the steps of the belief propagation (BP) algorithm. These RNN solutions weight the messages that are being passed as part of the BP method with fixed learnable weights.

In this work, we add compute to the message passing iterations, by turning the message graph into a graph neural network, in which one type of nodes, called variable nodes, processes the incoming messages with a FC network $g$. Since the space of possible messages is large and its underlying structure random, training such a network is challenging. Instead, we propose to make this network adaptive, by training a second network $f$ to predict the weights $\theta_g$ of network $g$.

This "hypernetwork" scheme, in which one network predicts the weights of another, allows us to control the capacity, e.g., we can have a different network per node or per group of nodes. Since the nodes in the decoding graph are naturally stratified and since a per-node capacity is too high for this problem, the second option is selected. Unfortunately, training such a hypernetwork still fails to produce the desired results, without applying two additional modifications. The first modification is to apply an absolute value to the input of network $f$, thus allowing it to focus on the confidence in each message rather than on the content of the messages. The second is to replace the $arctanh$ activation function that is employed by the check nodes with a high order Taylor approximation of this function, which avoids its asymptotes.

When applying learning solutions to algebraic block codes, the exponential size of the input space can be mitigated by ensuring that certain symmetry conditions are met. In this case, it is sufficient to train the network on a noisy version of the zero codeword. As we show, the architecture of the hypernetwork we employ is selected such that these conditions are met.

Applied to a wide variety of codes, our method outperforms the current learning based solutions, as well as the classical BP method, both for a finite number of iterations and at convergence of the message passing iterations.

## 2 Related Work

Over the past few years, deep learning techniques were applied to error correcting codes. This includes encoding, decoding, and even, as shown recently in [11], designing new feedback codes. The new feedback codes, which were designed by an RNN, outperform the well-known state of the art codes (Turbo, LDPC, Polar) for a Gaussian noise channel with feedback.

Fully connected neural networks were used for decoding polar codes [7]. For short polar codes, e.g., $n = 16$ bits, the obtained results are close to the optimal performance obtained with maximum a posteriori (MAP) decoding. Since the number of codewords is exponential in the number of information bits $k$, scaling the fully connected network to larger block codes is infeasible.

Several methods were introduced for decoding larger block codes ($n \geqslant 100$). For example in [17] the belief propagation (BP) decoding method is unfolded into a neural network in which weights are assigned to each variable edge. The same neural decoding technique was then extended to the min-sum algorithm, which is more hardware friendly [16]. In both cases, an improvement is shown in comparison to the baseline BP method.

Another approach was presented for decoding Polar codes [5]. The polar encoding graph is partitioned into sub-blocks, and the decoding is performed to each sub-block separately. In [12] an RNN decoding scheme is introduced for convolutional and Turbo codes, and shown to achieve close to the optimal performance, similar to the classical convolutional codes decoders Viterbi and BCJR.

Our work decodes block codes, such as LDPC, BCH, and Polar. The most relevant comparison is with [18], which improve upon [17]. A similar method was applied to Polar code in [21], and another related work on Polar codes [5] introduced a non-iterative and parallel decoder. Another contribution learns the nodes activations based on components from existing decoders (BF, GallagerB, MSA, SPA) [22]. In contrast, our method learns the node activations from scratch.

The term *hypernetworks* is used to refer to a framework in which a network $f$ is trained to predict the weights $\theta_g$ of another network $g$. Earlier work in the field [14, 20] learned weights of specific layers in the context of tasks that required a dynamic behavior. Fuller networks were trained to predict video frames and stereo views [10]. The term itself was coined in [8], which employed such meta-functions in the context of sequence modeling. A Bayesian formulation was introduced in a subsequent work [15]. The application of hyper networks as meta-learners in the context of few-shot learning was introduced in [2].

An application of hypernetworks for searching over the architecture space, where evaluation is done with predicted weights conditioned on the architecture, rather than performing gradient descent with that architecture was proposed in [4]. Recently, graph hypernetworks were introduced for searching over possible architectures [23]. Given an architecture, a graph hypernetwork that is conditioned on the graph of the architecture and shares its structure, generates the weights of the network with the given architecture. In our work, a non-graph network generates the weights of a graph network. To separate between the two approaches, we call our method hyper-graph-network and not graph hypernetwork.

## 3 Background

We consider codes with a block size of $n$ bits. It is defined by a binary generator matrix $G$ of size $k \times n$ and a binary parity check matrix $H$ of size $(n - k) \times n$.

The parity check matrix entails a Tanner graph, which has $n$ variable nodes and $(n - k)$ check nodes, see Fig. 1(a). The edges of the graph correspond to the on-bits in each column of the matrix $H$. For notational convenience, we assume that the degree of each variable node in the Tanner graph, i.e., the sum of each column of $H$, has a fixed value $d_v$.

The Tanner graph is unrolled into a Trellis graph. This graph starts with $n$ variable nodes and is then composed of two types of columns, variable columns and check columns. Variable columns consist of *variable processing units* and check columns consist of *check processing units*. $d_v$ variable processing units are associate with each received bit, and the number of processing units in the variable column is, therefore, $E = d_v n$. The check processing units are also directly linked to the edges of the Tanner graph, where each parity check corresponds to a row of $H$. Therefore, the check

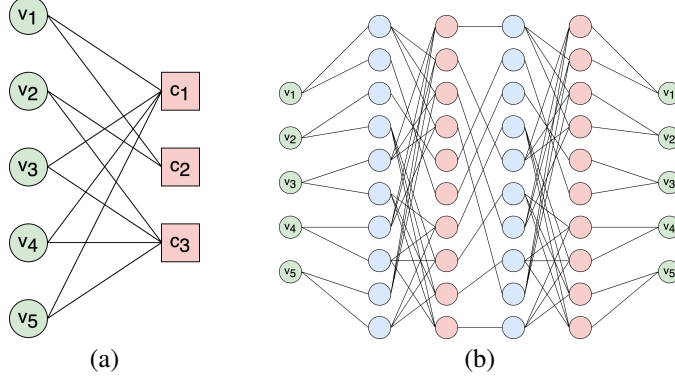

Figure 1: (a) The Tanner graph for a linear block code with $n = 5$, $k = 2$ and $d_v = 2$. (b) The corresponding Trellis graph, with two iteration.

columns also have $E$ processing units each. The Trellis graph ends with an output layer of $n$ variable nodes. See Fig. 1(b).

Message passing algorithms operate on the Trellis graph. The messages propagate from variable columns to check columns and from check columns to variable columns, in an iterative manner. The leftmost layer corresponds to a vector of log likelihood ratios (LLR) $l \in \mathbb{R}^n$ of the input bits:

$$l_v = \log \frac{\Pr\left(c_v = 1 | y_v\right)}{\Pr\left(c_v = 0 | y_v\right)},$$

where $v \in [n]$ is an index and $y_v$ is the channel output for the corresponding bit $c_v$, which we wish to recover.

Let $x^j$ be the vector of messages that a column in the Trellis graph propagates to the next column. At the first round of message passing $j = 1$, and similarly to other cases where $j$ is odd, a variable node type of computation is performed, in which the messages are added:

$$x_e^j = x_{(c,v)}^j = l_v + \sum_{e' \in N(v) \setminus \{(c,v)\}} x_{e'}^{j-1}, \tag{1}$$

where each variable node is indexed the edge $e = (c, v)$ on the Tanner graph and $N(v) = \{(c,v) | H(c,v) = 1\}$, i.e, the set of all edges in which $v$ participates. By definition $x^0 = 0$ and when $j = 1$ the messages are directly determined by the vector $l$.

For even $j$, the check layer performs the following computations:

$$x_e^j = x_{(c,v)}^j = 2 arctanh \left( \prod_{e' \in N(c) \setminus \{(c,v)\}} tanh \left( \frac{x_{e'}^{j-1}}{2} \right) \right) \tag{2}$$

where $N(c) = \{(c,v) | H(c,v) = 1\}$ is the set of edges in the Tanner graph in which row $c$ of the parity check matrix $H$ participates.

A slightly different formulation is provided by [18]. In this formulation, the $tanh$ activation is moved to the variable node processing units. In addition, a set of learned weights $w_e$ are added. Note that the learned weights are shared across all iterations $j$ of the Trellis graph.

$$x_e^j = x_{(c,v)}^j = \tanh \left( \frac{1}{2} \left( l_v + \sum_{e' \in N(v) \setminus \{(c,v)\}} w_{e'} x_{e'}^{j-1} \right) \right), \qquad \text{if } j \text{ is odd} \tag{3}$$

$$x_e^j = x_{(c,v)}^j = 2 arctanh \left( \prod_{e' \in N(c) \setminus \{(c,v)\}} x_{e'}^{j-1} \right) \qquad \text{if } j \text{ is even} \tag{4}$$

As mentioned, the computation graph alternates between variable columns and check columns, with $L$ layers of each type. The final layer marginalizes the messages from the last check layer with the logistic (sigmoid) activation function $\sigma$, and output $n$ bits. The $v$th bit output at layer $2L + 1$, in the weighted version, is given by:

$$o_v = \sigma \left( l_v + \sum_{e' \in N(v)} \bar{w}_{e'} x_{e'}^{2L} \right),$$ (5)

where $\bar{w}_{e'}$ is a second set of learnable weights.

## 4 Method

We suggest further adding learned components into the message passing algorithm. Specifically, we replace Eq. 3 (odd $j$) with the following equation:

$$x_e^j = x_{(c,v)}^j = g(l_v, x_{N(v,\backslash c)}^{j-1}, \theta_g^j),$$ (6)

where $x_{N(v,\backslash c)}^j$ is a vector of length $d_v - 1$ that contains the elements of $x^j$ that correspond to the indices $N(v) \setminus \{(c, v)\}$, and $\theta_g^j$ has the weights of network $g$ at iteration $j$.

In order to make $g$ adaptive to the current input messages at every variable node, we employ a hypernetwork scheme and use a network $f$ to determine its weights.

$$\theta_g^j = f(|x^{j-1}|, \theta_f)$$ (7)

where $\theta_f$ are the learned weights of network $f$. Note that $g$ is fixed to all variable nodes at the same column. We have also experimented with different weights per variable (further conditioning $g$ on the specific messages $x_{N(v,\backslash c)}^{j-1}$ for the variable with index $e = (v, c)$). However, the added capacity seems detrimental.

The adaptive nature of the hypernetwork allows the variable computation, for example to neglect part of the inputs of $g$, in case the input message $l$ contains errors.

Note that the messages $x^{j-1}$ are passed to $f$ in absolute value (Eq. 7). The absolute value of the messages is sometimes seen as measure for the correctness, and the sign of the message as the value (zero or one) of the corresponding bit [19]. Since we want the network $f$ to focus on the correctness of the message and not the information bits, we remove the signs.

The architecture of both $f$ and $g$ does not contain bias terms and employs the $tanh$ activations. The network $g$ has $p$ layers, i.e., $\theta_g = (W_1, ..., W_p)$, for some weight matrices $W_i$. The network $f$ ends with $p$ linear projections, each corresponding to one of the layers of network $g$. As noted above, if a set of symmetry conditions are met, then it is sufficient to learn to correct the zero codeword. The link between the architectural choices of the networks and the symmetry conditions is studied in Sec. 5.

Another modification is being done to the columns of the check variables in the Trellis graph. For even values of $j$, we employ the following computation, instead of Eq. 4.

$$x_e^j = x_{(c,v)}^j = 2 \sum_{m=0}^{q} \frac{1}{2m+1} \left( \prod_{e' \in N(c) \setminus \{(c,v)\}} x_{e'}^{j-1} \right)^{2m+1}$$ (8)

in which $arctanh$ is replaced with its Taylor approximation of degree $q$. The approximation is employed as a way to stabilize the training process. The $arctanh$ activation, has asymptotes in $x = 1, -1$, and training with it often explodes. Its Taylor approximation is a well-behaved polynomial, see Figure 2.

### 4.1 Training

In addition to observing the final output of the network, as given in Eq. 5, we consider the following marginalization for each iteration where $j$ is odd: $o_v^j = \sigma \left( l_v + \sum_{e' \in N(v)} \bar{w}_{e'} x_{e'}^j \right)$. Similarly to [18],

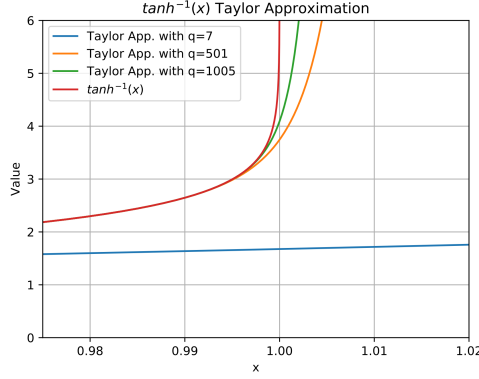

Figure 2: Taylor Approximation of the $arctanh$ activation function.

we employ the cross entropy loss function, which considers the error after every check node iteration out of the $L$ iterations:

$$\mathcal{L} = -\frac{1}{n} \sum_{h=0}^{L} \sum_{v=1}^{n} c_v \log(o_v^{2h+1}) + (1 - c_v) \log(1 - o_v^{2h+1}) \tag{9}$$

where $c_v$ is the ground truth bit. This loss simplifies, when learning the zero codeword, to $-\frac{1}{n} \sum_{h=0}^{L} \sum_{v=1}^{n} \log(1 - o_v^{2h+1})$.

The learning rate was $1e - 4$ for all type of codes, and the Adam optimizer [13] is used for training. The decoding network has ten layers which simulates $L = 5$ iterations of a modified BP algorithm.

## 5 Symmetry conditions

For block codes that maintain certain symmetry conditions, the decoding error is independent of the transmitted codeword [19, Lemma 4.92]. A direct implication is that we can train our network to decode only the zero codeword. Otherwise, training would need to be performed for all $2^k$ words. Note that training with the zero codeword should give the same results as training with all $2^k$ words.

There are two symmetry conditions.

1. For a check node with index $(c, v)$ at iteration $j$ and for any vector $b \in \{0, 1\}^{d_v - 1}$

$$\Phi\left(b^\top x_{N(\backslash v, c)}^{j-1}\right) = \left(\prod_{1}^{K} b_k\right) \Phi\left(x_{N(\backslash v, c)}^{j-1}\right) \tag{10}$$

where $x_{N(\backslash v, c)}^{j}$ is a vector of length $d_v - 1$ that contains the elements of $x^j$ that correspond to the indices $N(c) \setminus \{(c, v)\}$ and $\Phi$ is the activation function used, e.g., $arctanh$ or the truncated version of it.

2. For a variable node with index $(c, v)$ at iteration $j$, which performs computation $\Psi$

$$\Psi\left(-l_v, -x_{N(v, \backslash c)}^{j-1}\right) = -\Psi\left(l_v, x_{N(v, \backslash c)}^{j-1}\right) \tag{11}$$

In the proposed architecture, $\Psi$ is a FC neural network ($g$) with $tanh$ activations and no bias terms.

Our method, by design, maintains the symmetry condition on both the variable and the check nodes. This is verified in the following lemmas.

**Lemma 1.** *Assuming that the check node calculation is given by Eq. (8) then the proposed architecture satisfies the first symmetry condition.*

*Proof.* In our case the activation function $\Phi$ is Taylor approximation of $arctanh$. Let the input message at $j$ be $x^j_{N(\backslash v,c)} = \left(x^j_1, \ldots, x^j_K\right)$ for $K = d_v - 1$. We can verify that:

$$x^j(b_1 x^{j-1}_1, ..., b_K x^{j-1}_K) = 2 \sum_{m=0}^{q} \frac{1}{2m+1} (\prod_{k=1}^{K} b_k x^{j-1}_k)^{2m+1} = 2(\prod_{k=1}^{K} b_k) \sum_{m=0}^{q} \frac{1}{2m+1} (\prod_{k=1}^{K} x^{j-1}_k)^{2m+1}$$

$$= (\prod_{k=1}^{K} b_k) x_j(x^{j-1}_1, ..., x^{j-1}_K) \qquad \square$$

where the second equality holds since $2m + 1$ is odd.

**Lemma 2.** *Assuming that the variable node calculation is given by Eq. (6) and Eq. (7), g does not contain bias terms and employs the $tanh$ activation, then the proposed architecture satisfies the variable symmetry condition.*

*Proof.* Let $K = d_v - 1$ and $x^j_{N(v,\backslash c)} = \left(x^j_1, \ldots, x^j_K\right)$. In the proposed architecture for any odd $j \geqslant 0$, $\Psi$ is given as

$$g\left(l_v, x^{j-1}_1, \ldots, x^{j-1}_K, \theta^j_g\right) = tanh\left(W_p^\top \; ... \; tanh\left(W_2^\top tanh\left(W_1^\top\left(l_v, x^{j-1}_1, \ldots, x^{j-1}_K\right)\right)\right)\right)$$
(12)

where $p$ is the number of layers and the weights $W_1, ..., W_p$ constitute $\theta^j_g = f(|x^{j-1}|, \theta_f)$.

For real valued weights $\theta^{lhs}_g$ and $\theta^{rhs}_g$, since $tanh(x)$ is an odd function for any real value input, if $\theta^{lhs}_g = \theta^{rhs}_g$ then $g\left(l_v, x^{j-1}_1, \ldots, x^{j-1}_K, \theta^{lhs}_g\right) = -g\left(-l_v, -x^{j-1}_1, \ldots, -x^{j-1}_K, \theta^{rhs}_g\right)$. In our case, $\theta^{lhs}_g = f(|x^{j-1}|, \theta_f) = f(|-x^{j-1}|, \theta_f) = \theta^{rhs}_g$.

$\square$

# 6  Experiments

In order to evaluate our method, we train the proposed architecture with three classes of linear block codes: Low Density Parity Check (LDPC) codes [6], Polar codes [1] and Bose–Chaudhuri–Hocquenghem (BCH) codes [3]. All generator matrices and parity check matrices are taken from [9].

Training examples are generated as a zero codeword transmitted over an additive white Gaussian noise. For validation, we use the generator matrix $G$, in order to simulate valid codewords. Each training batch contains examples with different Signal-To-Noise (SNR) values.

The hyperparameters for each family of codes are determined by practical considerations. For Polar codes, which are denser than LDPC codes, we use a batch size of 90 examples. We train with SNR values of $1dB, 2dB, .., 6dB$, where from each SNR we present 15 examples per single batch. For BCH and LDPC codes, we train for SNR ranges of $1 - 8dB$ (120 samples per batch). In our results we report, the test error up to an SNR of $6dB$, since evaluating the statistics for higher SNRs in a reliable way requires the evaluation of a large number of test samples (recall that in train, we only need to train on a noisy version of a single codeword). However, for BCH codes, which are the focus of the current literature, we extend the tests to $8dB$ in some cases.

In our experiments, the order of the Taylor series of $arctanh$ is set to $q = 1005$. The network $f$ has four layers with 32 neurons at each layer. The network $g$ has two layer with 16 neurons at each layer. For BCH codes, we also tested a deeper configuration in which the network $f$ has four layers with 128 neurons at each layer.

The results are reported as bit error rates (BER) for different SNR values (dB). Fig. 3 shows the results for sample codes, and Tab. 1 lists results for more codes. As can be seen in the figure for Polar(128,96) code with five iteration of BP we get an improvement of $0.48dB$ over [18]. For LDPC MacKay(96,48) code, we get an improvement of $0.15dB$. For the BCH(63,51) with large $f$ we get

Table 1: A comparison of the negative natural logarithm of Bit Error Rate (BER) for three SNR values of our method with literature baselines. Higher is better.

| Method | BP | | | [18] | | | Ours | | | Ours deeper $f$ | | |
|---|---|---|---|---|---|---|---|---|---|---|---|---|
| | 4 | 5 | 6 | 4 | 5 | 6 | 4 | 5 | 6 | 4 | 5 | 6 |
| | colspan | | | | | — after five iterations — | | | | | | |
| Polar (63,32) | 3.52 | 4.04 | 4.48 | 4.14 | 5.32 | 6.67 | 4.25 | 5.49 | 7.02 | — | — | — |
| Polar (64,48) | 4.15 | 4.68 | 5.31 | 4.77 | 6.12 | 7.84 | 4.91 | 6.48 | 8.41 | — | — | — |
| Polar (128,64) | 3.38 | 3.80 | 4.15 | 3.73 | 4.78 | 5.87 | 3.89 | 5.18 | 6.94 | — | — | — |
| Polar (128,86) | 3.80 | 4.19 | 4.62 | 4.37 | 5.71 | 7.19 | 4.57 | 6.18 | 8.27 | — | — | — |
| Polar (128,96) | 3.99 | 4.41 | 4.78 | 4.56 | 5.98 | 7.53 | 4.73 | 6.39 | 8.57 | — | — | — |
| LDPC (49,24) | 5.30 | 7.28 | 9.88 | 5.49 | 7.44 | 10.47 | 5.76 | 7.90 | 11.17 | — | — | — |
| LDPC (121,60) | 4.82 | 7.21 | 10.87 | 5.12 | 7.97 | 12.22 | 5.22 | 8.29 | 13.00 | — | — | — |
| LDPC (121,70) | 5.88 | 8.76 | 13.04 | 6.27 | 9.44 | 13.47 | 6.39 | 9.81 | 14.04 | — | — | — |
| LDPC (121,80) | 6.66 | 9.82 | 13.98 | 6.97 | 10.47 | 14.86 | 6.95 | 10.68 | 15.80 | — | — | — |
| MacKay (96,48) | 6.84 | 9.40 | 12.57 | 7.04 | 9.67 | 12.75 | 7.19 | 10.02 | 13.16 | — | — | — |
| CCSDS (128,64) | 6.55 | 9.65 | 13.78 | 6.82 | 10.15 | 13.96 | 6.99 | 10.57 | 15.27 | — | — | — |
| BCH (31,16) | 4.63 | 5.88 | 7.60 | 4.74 | 6.25 | 8.00 | 5.05 | 6.64 | 8.80 | 4.96 | 6.63 | 8.80 |
| BCH (63,36) | 3.72 | 4.65 | 5.66 | 3.94 | 5.27 | 6.97 | 3.96 | 5.35 | 7.20 | 4.00 | 5.42 | 7.34 |
| BCH (63,45) | 4.08 | 4.96 | 6.07 | 4.37 | 5.78 | 7.67 | 4.48 | 6.07 | 8.45 | 4.41 | 5.91 | 7.91 |
| BCH (63,51) | 4.34 | 5.29 | 6.35 | 4.54 | 5.98 | 7.73 | 4.64 | 6.08 | 8.16 | 4.67 | 6.19 | 8.22 |
| | | | | | | — at convergence — | | | | | | |
| Polar (63,32) | 4.26 | 5.38 | 6.50 | 4.22 | 5.59 | 7.30 | 4.59 | 6.10 | 7.69 | — | — | — |
| Polar (64,48) | 4.74 | 5.94 | 7.42 | 4.70 | 5.93 | 7.55 | 4.92 | 6.44 | 8.39 | — | — | — |
| Polar (128,64) | 4.10 | 5.11 | 6.15 | 4.19 | 5.79 | 7.88 | 4.52 | 6.12 | 8.25 | — | — | — |
| Polar (128,86) | 4.49 | 5.65 | 6.97 | 4.58 | 6.31 | 8.65 | 4.95 | 6.84 | 9.28 | — | — | — |
| Polar (128,96) | 4.61 | 5.79 | 7.08 | 4.63 | 6.31 | 8.54 | 4.94 | 6.76 | 9.09 | — | — | — |
| LDPC (49,24) | 6.23 | 8.19 | 11.72 | 6.05 | 8.34 | 11.80 | 6.23 | 8.54 | 11.95 | — | — | — |
| MacKay (96,48) | 8.15 | 11.29 | 14.29 | 8.66 | 11.52 | 14.32 | 8.90 | 11.97 | 14.94 | — | — | — |
| BCH (63,36) | 4.03 | 5.42 | 7.26 | 4.15 | 5.73 | 7.88 | — | — | — | 4.29 | 5.91 | 8.01 |
| BCH (63,45) | 4.36 | 5.55 | 7.26 | 4.49 | 6.01 | 8.20 | — | — | — | 4.64 | 6.27 | 8.51 |
| BCH (63,51) | 4.58 | 5.82 | 7.42 | 4.64 | 6.21 | 8.21 | — | — | — | 4.80 | 6.44 | 8.58 |

an improvement of $0.45dB$ and with small $f$ we get a similar improvement of $0.43dB$. Furthermore, for every number of iterations, our method obtains better results then [18]. We can also observe that our method with 5 iteration achieve the same results as [18] with 50 iteration, for BCH(63,51) and Polar(128,96) codes. Similar improvements were also observe for other BCH and Polar codes. Fig. 3(e) provides experiments for large and non-regular LDPC codes - WARN$(384, 256)$ and TU-KL$(96, 48)$. As can be seen, our method improves the results, even in non-regular codes where the degree varies. Note that we learned just one hypernetwork $g$, which corresponds to the maximal degree and we discard irrelevant outputs for nodes with lower degrees. In Tab. 1 we present the negative natural logarithm of the BER. For the 15 block codes tested, our method get better results then the BP and [18] algorithms. This results stay true for the convergence point of the algorithms, i.e. when we run the algorithms with 50 iteration.

To evaluate the contribution of the various components of our method, we ran an ablation analysis. We compare (i) our complete method, (ii) a method in which the parameters of $g$ are fixed and $g$ receives and additional input of $|x^{j-1}|$, (iii) a similar method where the number of hidden units in $g$ was increased to have the same amount of parameters of $f$ and $g$ combined, (iv) a method in which $f$ receives the $x^{j-1}$ instead of the absolute value of it, (v) a variant of our method in which $arctanh$ replaces its Taylor approximation, and (vi) a similar method to the previous one, in which gradient clipping is used to prevent explosion. The results, reported in Tab. 2 demonstrate the advantage of our complete method. We can observe that without hypernetwork and without the absolute value in Eq. 7, the results degrade below those of [18]. We can also observe that for (ii), (iii) and (iv) the method reaches the same low quality performance. For (v) and (vi), the training process explodes and the performance is equal to a random guess. In (vi), we train our method while clipping the arctanh at multiple threshold values (TH $= 0.5, 1, 2, 4, 5$, applied to both the positive and negative

Table 2: Ablation analysis. The negative natural logarithm of BER results of our complete method are compared with alternative methods. Higher is better.

| Code | BCH (31,16) | | BCH (63,45) | | BCH (63,51) | |
|---|---|---|---|---|---|---|
| Variant/SNR | 4 | 6 | 4 | 6 | 4 | 6 |
| (i) Complete method | 4.96 | 8.80 | 4.41 | 7.91 | 4.67 | 8.22 |
| (ii) No hypernetwork | 2.94 | 3.85 | 3.54 | 4.76 | 3.83 | 5.18 |
| (iii) No hypernetwork, higher capacity | 2.94 | 3.85 | 3.54 | 4.76 | 3.83 | 5.18 |
| (iv) No abs in Eq. 7 | 2.86 | 3.99 | 3.55 | 4.77 | 3.84 | 5.20 |
| (v) Not truncating $arctanh$ | 0.69 | 0.69 | 0.69 | 0.69 | 0.69 | 0.69 |
| (vi) Gradient clipping | 0.69 | 0.69 | 0.69 | 0.69 | 0.69 | 0.69 |
| [18] | 4.74 | 8.00 | 3.97 | 7.10 | 4.54 | 7.73 |
| [18] with truncated $arctanh$ | 4.78 | 8.24 | 4.34 | 7.34 | 4.53 | 7.84 |

sides, multiple block codes BCH(31,16), BCH(63,45), BCH(63,51), LDPC (49,24), LDPC (121,80), POLAR(64,32), POLAR(128,96), $L = 5$ iterations). In all cases, the training exploded, similar to the no-threshold vanilla arctanh (v). In order to understand this, we observe the values when arctanh is applied at initialization for our method and for [17, 18]. In [17, 18], which are initialized to mimic the vanilla BP, the activations are such that the maximal arctanh value at initialization is 3.45. However in our case, in many of the units, the value explodes at infinity. Clipping does not help, since for any threshold value, the number of units that are above the threshold (and receive no gradient) is large. Since we employ hypernetworks, the weights $\theta_g^j$ of the network $g$ are dynamically determined by the network $f$ and vary between samples, making it challenging to control the activations $g$ produces. This highlights the critical importance of the Taylor approximation for the usage of hypernetworks in our setting. The table also shows that for most cases, the method of [18] slightly benefits from the usage of approximated $arctanh$.

## 7 Conclusions

We presents graph networks in which the weights are a function of the node's input, and demonstrate that this architecture provides the adaptive computation that is required in the case of decoding block codes. Training networks in this domain can be challenging and we present a method to avoid gradient explosion that seems more effective, in this case, than gradient clipping. By carefully designing our networks, important symmetry conditions are met and we can train efficiently. Our results go far beyond the current literature on learning block codes and we present results for a large number of codes from multiple code families.

**Acknowledgments**

We thank Sebastian Cammerer and Chieh-Fang Teng for the helpful discussion and providing code for deep polar decoder. The contribution of Eliya Nachmani is part of a Ph.D. thesis research conducted at Tel Aviv University.

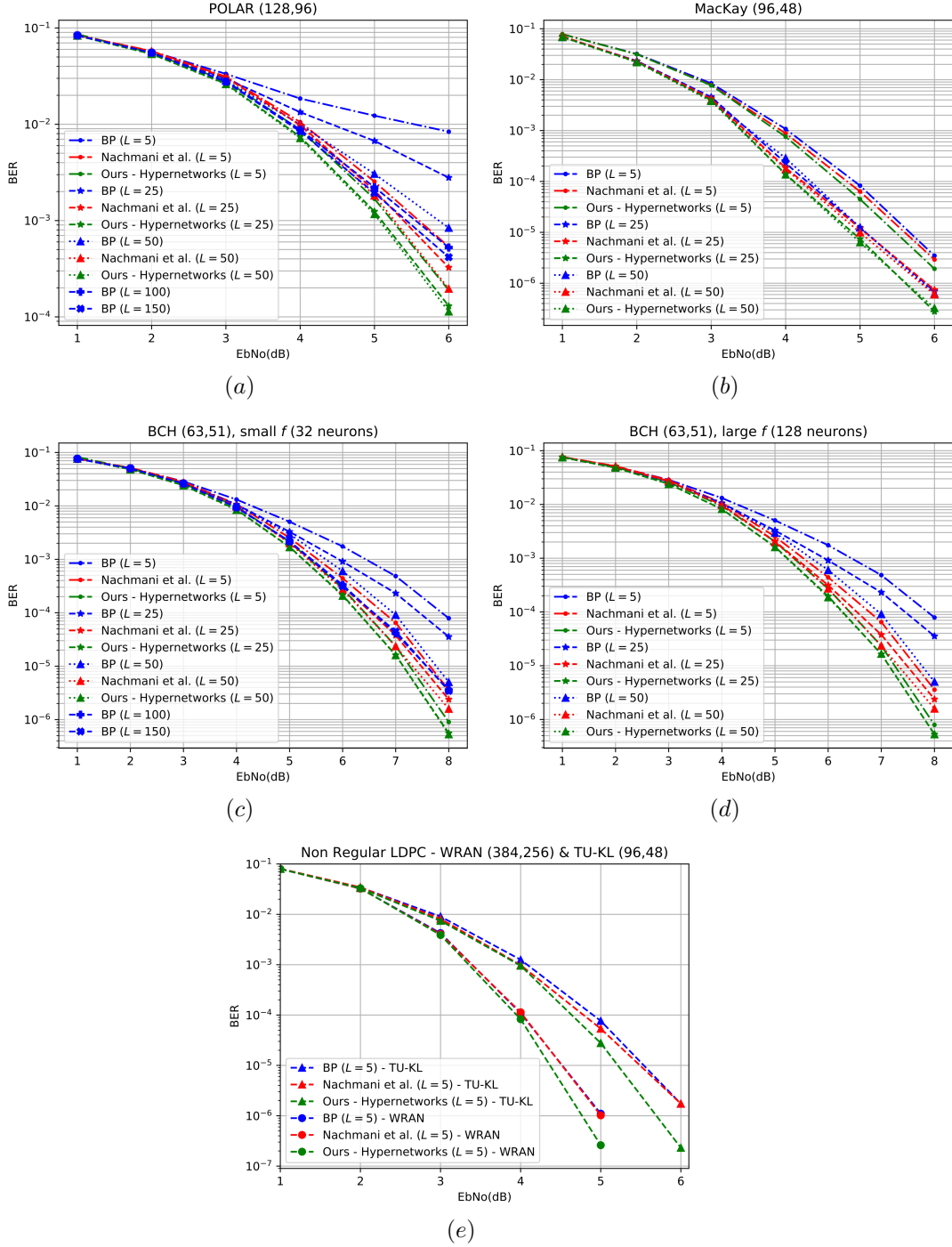

Figure 3: BER for various values of SNR for various codes. (a) Polar (128,96), (b) LDPC MacKay(96,48), (c) BCH (63,51), (d) BCH(63,51) with a deeper network $f$, (e) Large and non-regular LDPC codes: WRAN(384,256) and TU-KL(96,48).

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
