[Reviews · NeurIPS 2019]

Reviewer 1



In this paper, the authors propose to use a fully-connected NN to improve the BP decoding for block codes of regular degree distribution. The results are quite interesting because it shows that we can do better than BP for these regular codes by weighting the different contributions coming from the parity check. In a way, it tells each bit which parity check should trust more when doing each BP step and it allows the modified BP algorithm to converge faster and more accurately to the right code word. The gains are marginal, but given how good BP typically is that should not come as a surprise and should not be held against the paper. I have several comments about the paper that I would like to be addressed in the final version. The comment about changing the arctanh for a Taylor expansion is not a regularization of training, it is due to numerical instabilities and randomness of the stochastic gradient descent that can make intermediate x too close to one and making the arctanh to overflow. I can imagine thresholding the arctanh between -4 and 4 would have the same effect (and is easier to understand). Please say that you are doing the Taylor expansion to avoid numerical instabilities and I encourage you to apply the threshold of the arctanh. The authors should also address the issues of non-regular LDPC codes and how the training would be done in this case, as the f and g network will have to be trained for each potential degree. Would this hurt their performance? How good would it be compared to BP in this case, given that BP achieves capacity? I would also like to see in Figure 3 the results with BP with $L = \infty$, because clearly in 3 of the cases the BP needs more iterations while the proposed algorithm does not (only in (b) the BP has converged). The authors can complain that the BP needs more iterations, but BP is simpler than the proposed algorithm and we should know if just applying more BP iterations would be enough. Finally, it would also be interesting to see codes with thousands or tens of thousands of bits, because in that case the NNs would need to be larger and harder to train and BP will be closer to its ideal conditions. This would allow us to understand if the proposed technique is universally applicable or only is valid for short (or not-so-long) codes, which is still a relevant problem. Minor issues: In the introduction to Section 5, you give the impression that you might need to train the network for all possible codewords and that only training with the zero-word might be a way of cheating. I know this I not the case, but I would reconsider rephrasing. What happens if the network f is trained with x instead of |x|, given that you have a large enough network it should not matter. I know that After the feedback: 1 Thank you for changing the nonlinearity (and it is good that it did not work). It will be good to add this comment to the paper so people understand why the Taylor series expansion is needed. 2 The results for irregular codes it is also good because it shows that you can do this not only for regular codes and will make the paper stronger. 3 With larger codes (n = 1000 bits), the gain almost vanishes and if you had used 10^4 or 10^5, I would be surprised that you can measure the difference. This is expected and your algorithm would be more useful in the ten to hundred bits range. I have decided to raise my score on this paper.

Reviewer 2



While I acknowledge the novelties of the paper, as outlined above, I think the authors could have done a better job in presenting their results. Right now, the paper looks like a report of a series of tricks to perform better NN decoding of algebraic codes. What is the basic idea behind these tricks and why are they playing a major role in improving the performance with respect to previous methods, e.g. Nachmani et al.? I simply don't see an interesting framework behind this paper. Also, in the experimental results, why haven't the authors provided comparison with the state-of-the-art decoding methods? E.g. for polar codes, the methods should also be compared with the list-successive -cancellation decoder. Right now, the methods have a marginal improvement in themes of the error performance with respect to Nachmani's method.

Reviewer 3



The paper provides a new deep unfolding formulation of belief propagation. The idea looks natural and reasonable. However, I felt that the novelty is a bit weak because the idea is a straight extension of Nachmani's works. Numerical results show the new algorithm outperforms the conventional algorithm when the code is short BCH codes. The gain obtained by the modification is not so large (compared with Nachmani's original) but the idea deserves to be known. I have read the author response and my opinion remains the same.

[Author Response · NeurIPS 2019]



Figure 1: (a) Non-regular LDPC, (b) BP at convergence, (c) Large LDPC code, (d) Polar code with SCL

We thank our colleagues for their mostly supportive and very constructive and detailed feedback.

Reviewer #1: We will refer to the **Taylor approximation** in terms of numerical stability (line 6) and remove the
unfortunate term "regularize training" (l.134). Following the request, we tried to train our method while clipping the
arctanh at multiple threshold values (TH $= 0.5, 1, 2, 4, 5$, applied to both the positive and negative sides, multiple
block codes BCH(31,16), BCH(63,45), BCH(63,51), LDPC (49,24), LDPC (121,80), POLAR(64,32), POLAR(128,96),
$L = 5$ iterations). In all cases, the training exploded, similar to the no-threshold vanilla arctanh (l.209,214-215). In
order to understand this, we observe the values when arctanh is applied at initialization for our method and for [17,18].
In [17,18], which are initialized to mimic the vanilla BP, the activations are such that the maximal arctanh value at
initialization is 3.45. However in our case, in many of the units, the value explodes at infinity. Clipping does not help,
since for any threshold value, the number of units that are above the threshold (and receive no gradient) is large. Since
we employ hypernetworks, the weights $\theta_g^j$ of the network $g$ are dynamically determined by the network $f$ and vary
between samples, making it challenging to control the activations $g$ produces. This highlights the critical importance of
the Taylor approximation for the usage of hypernetworks in our setting. **Non-regular LDPC**: Following the request,
we ran experiments with WARN$(384, 256)$ and TU-KL$(96, 48)$. As can be seen in Fig. 1(a), our method improves
the results, even in non-regular codes where the degree varies. Note that we learned just one hypernetwork $g$, which
corresponds to the maximal degree and we discard irrelevant outputs for nodes with lower degrees. **BP at convergence**:
In the paper (l.203), we ran BP experiments until $L = 50$ considering it a convergence point. However, following
the review, we ran BP for many more iterations ($L = 150$, which is identical to $L = 100$, showing convergence) and
verified that our method indeed obtains considerably better performance. See Fig. 1(b) for a single result that reflects the
situation in all tested codes. **Large Codes:** Following the request, we tested WiMax$(1248, 1040)$, WiMax$(1056, 880)$
and WiMax$(1440, 1080)$. Fig. 1(c) shows an improvement for large SNR ($> 4dB$) despite (as R1 mentions) BP
being close to optimal in such lengths. **"Minor issues"**: We will clarify the sentence regarding training with the zero
codeword. The description of the second minor issue seems to have been cut in the middle and we are not sure that we
understand it. Running without the absolute value is part of the ablation.

Reviewer #2: The main idea behind the paper is that by employing decoders that are based on hypernetworks, we
are able to adapt dynamically to the received noisy codeword and improve the performance. However, applying
hypernetworks to such graph networks is uncharted territory, with many challenges that arise from the lack of control
over the weights at initialization, the dynamic nature of the weights, and the need to conform with specific symmetry
conditions (Lemmas 1,2). Please see the ablation analysis (l.205-216) and note that: (i) hypernetworks allow us to adapt
dynamically (l.120), (ii) abs(x) focuses on the reliability of the signal (l.122-125), (iii) Taylor approximation, please
see reply to R1, (iv) symmetry conditions, allow to train only on the zero codeword (l.146-148), (v) marginalization
at every iteration leads to a better gradient flow (l.140). **State of the art in polar codes**: In the paper, we incorporate
our method into the vanilla BP decoder. This decoder does not exploit the special structure of polar codes, leading to
results that are below the state of the art. As far as we know, no learning method obtains results on polar codes that are
better than the list-successive-cancellation (SCL) method. Following the review, we implemented our method on the BP
decoder by Arikan (E. Arikan, "Polar codes: A pipelined implementation"), which makes use of the structure of polar
codes. This is done by replacing the $f$ function in Arikan BP with a neural network $g$, whose weights are obtained from
another function $f$. The input to $f$ is the absolute value of the input LLRs. As Fig. 1(d) shows, our method improves
the Arikan BP decoding and is close to the performance of an SCL decoder, which is very close to the ML bound.

Reviewer #4: Thank you for pointing "Learning to decode LDPC codes with finite-alphabet message passing" by Vasic
et al, which we will cite. In their work, they learn the node activation based on components from existing decoders (BF,
GallagerB, MSA, SPA), while we learn the node activations from scratch. They publish results on Tanner(155,64) and
QC-LDPC(1296,72). In both codes, our performance is better across all SNR. For example, for the first code, for both
SNR=5.5 and SNR=6, we obtain a third of their bit error rate ($5 \times 10^{-7}$ and $7 \times 10^{-8}$, respectively). For the second
code, for SNR=2.5 (SNR=3), we obtain half of their bit error rate $1.3 \times 10^{-2}$ ($4.5 \times 10^{-3}$). **Additional codes**: See
comments to R1 regarding non-regular LDPC codes and large codes.

[Meta-Review · NeurIPS 2019]

This paper proposes a neural-network-based decoder architecture binary linear block codes with constant-degree variable nodes. It is based on message passing on the unfolded Tanner graph but replaces the variable-node operation in each iteration with a neural network g, whose parameters are provided by another neural network f which takes the absolute values of the messages as its input. (The check-node operation is also approximated via the Taylor expansion of arctanh as in (8).) Experimental results are provided to demonstrate that the proposed scheme performs well for various different types of codes. Although the review scores were around the acceptance threshold in the initial round of review, after the authors' rebuttal two reviewers have raised their scores, so that now all the reviewers are positive. I would thus like to recommend acceptance of this paper. Minor point: Line 81: has a fixed value(d)